# Relative Water Content, Proline, and Antioxidant Enzymes in Leaves of Long Shelf-Life Tomatoes under Drought Stress and Rewatering

**DOI:** 10.3390/plants11223045

**Published:** 2022-11-10

**Authors:** Cristina Patanè, Salvatore L. Cosentino, Daniela Romano, Stefania Toscano

**Affiliations:** 1CNR-Istituto per la BioEconomia (IBE), Sede Secondaria di Catania, Via P. Gaifami 18, 95126 Catania, Italy; 2Dipartimento di Agricoltura, Alimentazione e Ambiente, Università degli Studi di Catania, via Valdisavoia 5, 95123 Catania, Italy

**Keywords:** *Solanum lycopersicum* L., water stress, local landraces, malondialdehyde (MDA), proline, rewatering

## Abstract

Some physiological, oxidative, and antioxidant enzymatic patterns were assessed in plants of three local Sicilian landraces of long shelf-life tomatoes (‘Custonaci’, ‘Salina’, and ‘Vulcano’), as compared to the commercial tomato hybrid ‘Faino’ (control). Three water treatments were considered in open-field: DRY (no irrigation); IRR (long-season full irrigation); REW (drought/rewatering cycles). During the growing season, soil water deficit (SWD) was estimated, and relative water content (RWC), specific leaf area (SLA), proline and malondialdehyde (MDA) content, and glutathione peroxidase (GPX), catalase (CAT), and superoxide dismutase (SOD) activities were measured in leaves. Differently from control, RWC in local landraces exhibited a similar pattern in REW and DRY, indicating a low capacity to re-hydrate after rewatering. Positive correlation of proline content vs. SWD in all local landraces highlights an osmotic adjustment occurring in these tomatoes in response to limited soil water content. Long shelf-life tomatoes suffered minor oxidative stress due to severe soil water deficit, as revealed by the lower levels of MDA with respect to the control. Significant correlation of CAT vs. SWD for all tomatoes indicates that this antioxidant enzyme, among those analyzed, may be considered as a biomarker for a water stress condition more than for oxidative stress due to water deficit.

## 1. Introduction

Drought stress is a major abiotic stress that adversely affects crop growth and development and final yield [1]. A poor water content in plants due to severe soil water deficit may determine an imbalance between production and scavenging of reactive oxygen species (ROS) such as hydrogen peroxide (H_2_O_2_) or superoxide anion (O^2−^) [2]. Notably, stress-induced excessive production of ROS causes lipid peroxidation and malondialdehyde (MDA) accumulation, which lead to cell membrane damages and, ultimately, to cell death [3]. In this sense, MDA is assumed as a good indicator of cell membrane stability under stress [4].

During oxidative stress, plants are able to counteract the excessive production of ROS by means of an antioxidant response [4]. Antioxidants, such as catalase (CAT), glutathione peroxidase (GPX), and superoxide dismutase (SOD), are produced by plants as protectors against oxidative damages due to a range of environmental factors such as drought, salinity, and other environmental stressors. These factors of stress are typical of semi-arid regions [5], and their adverse effects upon crops are increasingly severe due to global climate warming.

Among a number of physiological mechanisms developed by plants to counteract the adverse effects of drought and salt stress, osmoregulators, such as proline, are accumulated by plants to maintain the osmotic balance (osmotic adjustment) under a stressful environment, without damaging cellular activities [6]. It has been suggested that plants with increased proline concentration may better tolerate these stresses, improving their growth under such limiting environmental conditions. In this regard, proline may be considered as a key non-enzymatic antioxidant against stress-induced ROS and, thus, a good indicator of water stress tolerance [7].

Tomato (*Solanum lycopersicum* L.) is an important horticultural crop widely spread all over the world. However, water stress, with flowering stage being the most sensitive to drought, may adversely affect its growth and productivity, leading to final low fruit yields [8]. Among several types of tomato, the local Mediterranean landraces of long shelf-life tomato are traditionally cultivated under rain-fed conditions in the semi-arid regions of South Italy [9]. These tomatoes, as observed in other tomato landraces recovered throughout the Mediterranean basin [2], have developed several physiological and biochemical mechanisms that alleviate the effects of drought stress [2,9]. Therefore, these tomatoes can be considered as a good option for those semi-arid cultivation areas unsuitable to water-demanding commercial tomatoes. Local landraces of long shelf-life tomato also produce fruits with high sensory and nutritional properties [10].

In recent years, these tomatoes have been extensively studied for their physiological, biochemical, and fruit quality response to severe soil water deficit conditions under open- field conditions [9,11,12,13,14]. A more recent study revealed that rewatering after long-lasting periods of drought in long shelf-life tomato may be beneficial in terms of crop productivity and may help the fruits to retain high levels of nutritional quality, when compared to overstressing conditions under no water supply [14]. However, no literature exists on the plant response to drought induced by the application of repeated drought and rewatering, in terms of oxidative stress and antioxidant enzymatic activity, in long shelf-life tomato under open-field conditions.

Keeping in view the increasing drought stress concerns upon tomato crop productivity, the present study was conducted to investigate some physiological, oxidative, and antioxidant enzymatic patterns in plants of three long shelf-life tomatoes exposed to prolonged drought or rehydration after long drought periods in a semi-arid climate. The final goal of this study was the identification of tolerant genotypes with beneficial stress-related traits, to be adopted in marginal lands or to be introduced in breeding programs for the development of tomato genotypes able to cope with actual and future climatic changes.

## 2. Results

### 2.1. Meteorological Course

The meteorological course during the field experiment was that of a typically Mediterranean environment, with a hot and dry summer (Figure 1). Maximum temperature measured during the crop growing season ranged between 18.4 °C (early May) and 36.0 °C (mid-July), and minimum between 10.0 °C (early May) and 22.2 °C (early July). Total rainfall during the experiment was very scarce (total 26.0 mm from early May to late July). Maximum daily reference evapotranspiration (ET_0_) (approx. 8 mm) during the experiment was recorded in mid-July, according to the highest maximum air temperature.

### 2.2. Course of SWD

The course of soil water deficit as measured throughout the growing season indicates that soil water content in the DRY treatment declined just after the cut of irrigation, dropping below the theoretical threshold for irrigation (SWD 66%) from early June onwards, and approaching the wilting point (SWD ≥ 80%) in late July (Figure 2).

In the IRR treatment, the available soil water content widely fluctuated, according to the time of irrigation. Since the volume of water supplied by irrigation was calculated indirectly, on the basis of cumulative ETc, and not on the basis of actual soil water content at the time of irrigation, often irrigation in IRR was applied before or after reaching the 66% SWD threshold. Moreover, in ‘Faino Hy’, irrigation in IRR sometimes was not adequate to fulfil the soil to field capacity, therefore leading to a soil water deficit that may have determined a water stress condition even in well-irrigated plots.

In REW, the course of SWD overlapped that of DRY, until the first rewatering (June 23) to restore soil water content back to field capacity (SWD 0%); then it dropped again, reaching wilting point (or values lower than it) in a few days, until the second rewatering in July. However, as reported for IRR, the second irrigation in REW was not adequate to fill soil water content up to field capacity. This was particularly true in ‘Salina’, where SWD overpassed the 66% threshold already in late May and dropped down to 100% in early June; therefore, the amount of water to distribute with irrigation in REW was underestimated in both rewaterings.

### 2.3. RWC of Leaves

Results for RWC measurements during the growing season are reported in Table 1 and Figure 3. RWC was kept significantly higher in IRR (*I*, *p* ≤ 0.001, Table 1), approaching 90% in mid-June, and declining later on.

In REW, except for a few cases, RWC exhibited trends and values similar to those in DRY, despite the two rewaterings. However, values of RWC always >75% revealed a good plant water status even under no irrigation in DRY. The highest RWC values were measured in the control ‘Faino Hy’ (*G*, *p* ≤ 0.01). Plants in local tomatoes exhibited a lower capacity to re-hydrate after rewatering, as compared to ‘Faino Hy’ (*I* × *G*, *p* ≤ 0.05). In fact, this last genotype showed a different pattern of RWC in REW, as compared to long shelf-life tomatoes, since its leaves well rehydrated after the first rewatering, and increased their RWC constantly up to the second rewatering. A similar pattern was observed in ‘Vulcano’; however, its RWC in REW was kept more or less stable throughout the growing season. In ‘Custonaci’ and ‘Salina’, RWC was more or less stable throughout the crop season.

### 2.4. Course of SLA

Specific leaf area was measured from values of leaf area and dry weight of the same leaves sampled for RWC measurements. In all tomatoes, SLA exhibited an increasing trend until approximately half-June, i.e., during the vegetative period until full flowering (*T*, *p* ≤ 0.001). Indeed, during this period, plant growth and leaf expansion are maximized. After that, SLA started declining in all water treatments (Figure 4). Among genotypes, ‘Custonaci’ exhibited the lowest values of SLA (*G*, *p* ≤ 0.001). Across genotypes and times of measurement, significantly lower values of SLA were measured in DRY (*I*, *p* ≤ 0.001). However, the differences among water treatments became clear only at the end of the growing season (from mid-July onwards), when values of SLA always lower than in IRR were measured in DRY. This pattern was more evident in ‘Vulcano’ (*I* × *G* × *T*, *p* ≤ 0.001). Differently, in the control ‘Faino Hy’, SLA values of the three treatments overlapped throughout the whole growing season.

### 2.5. Course of Proline

The time-course of proline content in leaves is reported in Table 1 and Figure 5. Among tomatoes, overall, significantly less proline was accumulated in leaves of ‘Vulcano’ and ‘Faino Hy’ during the growing season in all water treatments (*G*, *p* ≤ 0.001).

From an overall overview, it is evident that the content of proline was kept constantly low (<5 µmol g^−1^) under high rates of irrigation in IRR in all tomatoes (*I*, *p* ≤ 0.001). Only in ‘Faino Hy’, the differences in proline content among water treatments up to the first rewatering in REW were not so clear. In ‘Custonaci’, proline content originally exhibited the same pattern in DRY and REW, but after each rewatering the content in REW decreased, approaching the low content in IRR. Similarly, in ‘Salina’ the proline content did not differ between DRY and REW (>6 µmol g^−1^), but in this last, after a rapid decline following the first rewatering, proline content quickly rose, again to levels close to those in DRY. Plants of this tomato did not respond to the second rewatering in REW in terms of proline, which was kept high despite irrigation. A similar course was observed in ‘Vulcano’, although the overall differences among treatments were minor. In the control ‘Faino Hy’, proline content after the first irrigation in REW decreased and was maintained close to that of IRR (<4 µmol g^−1^) afterwards.

### 2.6. Course of Malondialdehyde (MDA)

Overall, malondialdehyde (MDA) tended to increase with time (*T*, *p* ≤ 0.001) (Table 2). Its content was significantly lower at high rates of irrigation (*I*, *p* ≤ 0.001). Among tomatoes, ‘Faino Hy’ exhibited the greatest MDA contents (*G*, *p* ≤ 0.001) across the irrigation treatments and the times of measurement. MDA measured in local tomatoes did not much vary with the irrigation treatment, always maintaining values more or less constantly below 6 µmol g^−1^ of FW (Figure 6) (*I × G*, *p* ≤ 0.001). Instead, MDA exhibited a noticeable increase in the commercial tomato, starting from late June onward in all water treatments up to a final content that approached (in REW) or exceeded (in DRY) 8 µmol g^−1^. Moreover, differently than observed in local types, MDA in ‘Faino Hy’ was maintained constantly higher in DRY from the first rewatering in REW (after late June) onwards (*I* × *G* × *T*, *p* ≤ 0.001).

All tomatoes responded to rewaterings (more clearly after the first one, i.e., that of late June), with a clear but temporary decline in MDA of leaves that, however, was not maintained afterwards. 

### 2.7. Course of Antioxidant Enzymes (GPX, CAT, SOD)

The glutathione peroxidase (GPX) activity was monitored during the growing season in leaves of the four tomatoes under the three water treatments (Figure 7). GPX was the highest under severe soil water deficit conditions (DRY) and the lowest under full irrigation (IRR) (*I*, *p* ≤ 0.001). Among tomatoes, greater levels of GPX activity were measured in ‘Faino Hy’ (*G*, *p* ≤ 0.001). In local tomatoes, GPX was quite similar in the three water treatments until mid-July, keeping values quite low (<8 µmol min^−1^ g^−1^ FW). A week later (i.e., at the last measurement), GPX sharply increased at greater rates in REW and more in DRY. In the commercial tomato, where overall the enzyme activity was greater, a clear rise in GPX activity occurred a week earlier, with values >15 µmol min^−1^ g^−1^ FW in DRY, and at the last measurement, GPX exceeded 16 µmol min^−1^ g^−1^ FW even in IRR (*I × G × T*, *p* ≤ 0.001).

Catalase (CAT) exhibited a different trend when compared to GPX (Figure 8). CAT activity was the highest in ‘Custonaci’ (*G*, *p* ≤ 0.001) (Table 2). Among water treatments, greater CAT activity was measured under DRY (*G, p* ≤ 0.001). Differences for this enzyme among water treatments were clear already from the early measurements. Overall, CAT was lower in IRR, with differences from DRY and REW which were already evident from the first irrigations in IRR in ‘Salina’ (*I* × *G* × *T*, *p* ≤ 0.001). In turn, CAT activity was maximized under DRY, peaking at 0.14 nmol min^−1^ g^−1^ FW in late June, and again in ‘Salina’. CAT seemed to be sensitive to rewatering, decreasing after irrigation in REW, although to a different extent depending on cultivar. In ‘Vulcano’ and the commercial ‘Faino Hy’, CAT activity in REW promptly declined just after each of the two irrigations, to increase again few days later. This course in REW was also observed in ‘Custonaci’, although CAT decrease was evident only after the second irrigation.

Superoxide dismutase activity (SOD) was monitored along with the other antioxidant enzymes in all tomatoes during the growing season (Figure 9). All tomatoes behaved differently in terms of SOD activity, with ‘Salina’ and ‘Vulcano’ exhibiting, respectively, the lowest and the highest activity (*G*, *p* ≤ 0.001) (Table 2). Water regime, as well, significantly affected SOD activity, which was the highest under DRY (*I*, *p* ≤ 0.001).

All tomatoes responded differently to water regimes during the growing season in terms of SOD activity (*I* × *G* × *T*, ≤ 0.001). In ‘Custonaci, after an initial slight increase in all water treatments, SOD progressively declined from early June onwards in IRR, but steeply increased (exceeding 60 U mg^−1^ FW) in the two water-stressed treatments (REW and DRY), decreasing afterwards under both irrigations, more clearly in REW, probably as a (delayed) response to rewatering. In ‘Salina’, the differences among water treatments were negligible up to late June, but later on, SOD activity increased in DRY, while in REW and IRR it still declined until the last measurement (late July). SOD in ‘Vulcano’ exhibited a trend similar to that described in ‘Custonaci’, but the enzyme activity was kept always lower in IRR even in the early measurements, never exceeding 40 U mg^−1^. In the control ‘Faino Hy’, SOD showed a similar course in all water treatments, peaking in mid-June (i.e., earlier than local types) (up to 58 U mg^−1^ in DRY), and declining afterwards, more steeply under no irrigation.

### 2.8. Fruit Yield per Plant

Fruit yield per plant was measured at the end of growing season. All tomato plants yielded significantly less when not irrigated (DRY), providing a final production that was 38% lower than that obtained under well-watered conditions (IRR) (Table 3).

As expected, across water regimes, the commercial ‘Faino Hy’ produced significantly more than local landraces; among these last, ‘Salina’ was the most productive and ‘Vulcano’ was the least productive. Significant *I* × *G* interaction (*p* ≤ 0.01) revealed a different sensitivity to restricted soil water availability among genotypes. More precisely, ‘Custonaci’ and ‘Salina’ produced 34% and 44% less, respectively, moving from IRR to DRY, revealing the greatest and the lowest water-stress tolerance. The yield response to water shortage in REW was also genotype-dependent. Indeed, all tomatoes positively reacted to rewaterings, raising their productivity (+28% on average) over the DRY control; however, the yield increase was maximized in ‘Salina’ (+39%) and minimized in the control ‘Faino Hy’ (+25%).

### 2.9. Relationships

Relationships among the different traits measured throughout the growing season in the four tomatoes were studied separately per cultivar (Table 4, Table 5, Table 6 and Table 7).

Leaf proline content was positively correlated to SWD in all local landraces (0.05 ≥ *p ≤* 0.001) but not in the commercial tomato. Neither RWC nor SLA correlated significantly with SWD (*p* > 0.05). Instead, when data of RWC were plotted vs. those of SLA, significant positive relationships were described in all local tomatoes (0.05 ≥ *p ≤* 0.01). However, the relationship was not significant in the commercial type (*p* > 0.05).

No significant correlation of MDA was found with SWD and proline. Interestingly, MDA correlated positively with CAT (*p* ≤ 0.05), GPX (*p* ≤ 0.001), and RWC (*p* ≤ 0.05) in the commercial tomato (‘Faino Hy’) but not in local landraces.

When SWD was plotted vs. all the antioxidant enzymes, significant positive correlations with CAT activity were found in all tomatoes (*p* ≤ 0.001), which indicate that this enzyme activity in tomato leaves is largely influenced by the soil water content at the time of measurement. The same did not occur for GPX, whose significant correlation (r = 0.53 *) was found only in ‘Salina’ of long storage tomato, revealing a minor influence of soil water deficit on this enzyme activity.

CAT activity values were also plotted against RWC measured at the same time, highlighting a slight but significant inverse correlation (r = −0.50 *), again in ‘Salina’ of local tomato. However, no significant relationship was described for CAT vs. RWC in the other tomatoes (*p* > 0.05).

SOD was found to correlate significantly and positively only with CAT and in ‘Custonaci’ (r = 0.52 *).

## 3. Discussion

In this study, the effects of severe drought conditions on some physiological and metabolic traits were examined in leaves of three landraces of long shelf-life tomatoes in a semi-arid environment, as compared with a commercial cultivar of mini plum tomato. The water stress conditions were imposed via no irrigation following plant establishment (DRY treatment), or by long-lasting drought periods followed by rewaterings (REW treatment). A fully irrigated control (IRR) was also considered for all tomatoes.

Relative water content (RWC) of leaves indicates the level of leaf hydration with respect to the maximum (the greater the RWC, the higher the hydration level) [9]. In this experiment, as expected, RWC was maximized under frequent irrigations in IRR. Differently than commercial tomato, RWC in local landraces exhibited a similar pattern in REW and DRY, indicating a low capacity to re-hydrate after rewaterings in REW. However, minor differences were ascertained among genotypes of long shelf-life tomato. According to Patanè et al. [9], this trait is not associated with the geographical area of recovery of local landraces, ‘Vulcano’ and ‘Salina’ both being from the Eolian archipelago. Low responsiveness to rewatering in terms of rehydration may in turn reveal a greater dehydration tolerance occurring in local landraces, i.e., greater plant aptitude to maintain normal functions under low tissue RWC [15]. No relationship of RWC vs. soil water deficit (SWD) was found, which may be associated with the time of soil water content measurement (just after irrigation in IRR): soil water content promptly changes after irrigation, however, plants take longer to respond to water applied.

Specific leaf area (SLA), which indicates the thickness of leaves, was rather similar in all tomato genotypes, including the control, and all water treatments. Values of SLA measured at the end of the growing season in almost all local landraces were lower in DRY than in IRR, revealing a leaf blade tendency to thicken under prolonged soil water deficit. This result may be considered as an adaptive mechanism of ‘osmotic adjustment’ occurring overall in these landraces that, under restricted soil water content, reduces leaf expansion, maintaining leaf dry matter [16]. Positive relationships of SLA vs. RWC described in all local tomatoes indicate that these types modulate their leaf growth according to leaf turgor (the higher the RWC, the lower the leaf thickness). The lack of relationship of SLA vs. SWD may be ascribed to the same assumptions above reported for RWC.

Proline is an osmolyte that has been reported to accumulate in leaves or increase in its concentration in response to abiotic stresses, including water stress, to protect cell tissues from oxidative damage [17]. Its role in water-stressed plants, however, is debated, because some authors reported that proline accumulates in plants as a symptom rather than an adaptive mechanism to stress [18,19]. In this experiment, overall proline content was kept low under frequent irrigations (IRR) and high under no irrigation (DRY). Moreover, a positive correlation of proline content vs. SWD was described in all local landraces but not in the commercial ‘Faino Hy’. These results highlight an osmotic adjustment occurring in plants of long shelf-life in response to limited soil water content. Similar findings were reported in a pool of long shelf-life genotypes cultivated in the same semi-arid environment of the present research [9]. The authors suggested that the rise in leaf proline concentration in these landraces could be considered as an adaptive mechanism deriving from a prolonged exposure to soil water deficit rather than a symptom of a water-stressed condition. Similarly, Desoky et al. [20], working on maize cultivated under well-watered or drought conditions, demonstrated that an increased proline biosynthesis in plant cells enhances the plant defense system to avoid oxidative damage stimulated by water stress.

An inverse relationship of proline vs. RWC has been described for long shelf-life tomatoes grown under dry conditions in a Mediterranean semi-arid environment [9]. This inverse relationship changed from linear to exponential for values of RWC <72%, indicating a better aptitude of tissue cells to osmoregulate at reduced water potential (higher rates of proline increase at lower RWC). In our experiment, a tendency of proline content to increase exponentially at lower RWC was also observed (data not shown). However, the relationship was not so clear, probably since RWC never dropped below 74%.

Along with proline, malondialdehyde (MDA) content was measured in tomato leaves during the growing season. Notably, malondialdehyde (MDA) is one of the final products of polyunsaturated fatty acids peroxidation in cells, whose amount denotes the extent of a damage occurring in tomato plants exposed to water stress [2]. In the present experiment, overall, progressive drought conditions did not induce additional oxidative stress in local landraces of tomato compared to well-watered conditions, as revealed by the levels of MDA, similar in all the irrigation treatments. Particularly low levels of MDA in ‘Vulcano’ (long shelf-life tomato), even in DRY, denote a great resistance to oxidative stress. Differently, a great responsiveness to prolonged drought conditions was observed in the commercial tomato, whose levels of MDA steeply increased from June onward, more clearly in DRY. In this tomato, rewatering seemed to mitigate the oxidative stress due to drought, the levels of MDA measured in REW being lower than those in DRY. Raziq et al. [4] noticed how a progressive stress (salt stress, in this case), induced rising levels of MDA in tomato seedlings, which in turn, caused a progressive decrease in membrane stability. In this regard, MDA has been proposed as a biomarker for oxidative stress under abiotic stress [21]. Our findings somehow indicate that long shelf-life tomatoes, ‘Vulcano’ in particular, suffered a minor oxidative stress due to severe soil water deficit. Raziq et al. [4] also suggested the occurrence of several strategies, such as antioxidative scavenging of ROS, adopted in tomato plants to counteract oxidative stress.

In this experiment, water-stress-induced activities of GPX, CAT, and SOD was analyzed in all tomatoes. GPX, CAT, and SOD are enzymatic antioxidants that counterbalance the deteriorative effects of ROS through their detoxification [22,23,24,25]. All the antioxidant enzymes were kept higher under drought-induced stress conditions. A similar promoting effect of soil water deficit upon SOD and CAT activity, associated with an increased MDA content, were reported in tomato, maize, and faba bean. Indeed, a cut of irrigation from 100% to 60% of soil field capacity led to a +100% increase in SOD and a +67% increase in CAT, in tomato [26], and a +55% increase in SOD and a +60% increase in CAT, in faba bean [23]. Similarly, a rise in SOD and CAT (+200%) was found in maize when moving irrigation from well-watered conditions to moderate or severe drought [20]. However, in the present study, CAT was the only enzyme to be positively correlated to SWD at the time of measurement, and responded promptly to rewaterings, decreasing soon after irrigation in REW. Lower ROS production under full irrigation may decrease the oxidative stress, leading to a reduced antioxidant production. Such reduced oxidative pressure corresponds well to a smaller CAT activity [22]. Lower levels of MDA in long shelf-life tomatoes alongside CAT levels similar to those of the commercial tomato denote a higher capacity to tolerate long-lasting drought stress for survival.

SOD is another important antioxidant enzyme that promotes the dissociation of superoxide anion into molecular oxygen and hydrogen peroxide; this last is then detoxified by other antioxidant enzymes (such as CAT) to water [22,23]. In this experiment, soil water availability induced SOD patterns which differed between tomato genotypes, with no differences among water treatments; in ‘Salina’ (when excluding the last measurements) SOD levels were always higher under REW and DRY than in ‘Vulcano’.

Despite high levels of MDA having been demonstrated to denote cell membrane damage with severe consequences for plant growth and development [4], in the present experiment higher levels of MDA did not alter plant productivity in the commercial tomato ‘Faino Hy’, whose yield losses in DRY on IRR were consistent with those of local types.

However, if we consider their traditional method of cultivation (under no irrigation) [9], greater yield increases occurred in local tomatoes moving from DRY to REW conditions, compared to the commercial type. Therefore, water stress management through rehydration after long periods of drought may be suggested to enhance plant productivity in these landraces. These results may represent an important goal for long shelf-life tomatoes, keeping in view that they have never been included in breeding programs for improved productivity [10]. Repeated cycles of dehydration and rehydration in these tomatoes have also been reported to contribute to maintain high levels of antioxidants in fruits [14].

## 4. Materials and Methods

### 4.1. Open-Field Experiment

The field experiment was conducted during the summer season of 2016, in a flat site of the Eastern coast of Sicily (South Italy, 10 m a.s.l., 37°24′35.8” N Lat, 15°03′31.7” E Long), on a vertic xerochrepts soil. The soil had the following characteristics: clay 28.3%, sand 49.3%, loam 22.4%, organic matter 1.4%, pH 8.6, total N 1.0‰, available P_2_O_5_ 5 ppm, exchangeable K_2_O 245 ppm, bulk density 1.1 g cm^−3^, field capacity (−0.03 MPa) 0.27 g g^−1^, wilting point (−1.5 MPa) 0.11 g g^−1^.

Three local Sicilian landraces of long-shelf-life tomato (’Custonaci’, from the western coast of Sicily, ‘Salina’ and ‘Vulcano’, from Eolian islands) belonging to the germplasm collection at CNR-IBE (Catania, Italy) were investigated in this experiment and compared to the commercial ‘Faino Hy’ (Syngenta seeds, The Netherland) of mini plum.

Plants were transplanted in the field on 4 May, at the 4-leaf stage. Single plots measured 12.0 m^2^ (3.0 × 4.0 m). Plant spacing was 0.75 m between rows and 0.40 m within rows, leading to a 3.3 plants m^−2^ plant density. Before transplanting, 75, 100, and 100 kg ha^−1^ of N (as ammonium sulphate), P_2_O_5_ (as mineral perphosphate), and K_2_O (as potassium sulphate), respectively, were distributed in field. Approximately 30 days after transplanting, a further 75 kg ha^−1^ of N (as ammonium nitrate) was supplied as top dressing.

Two experimental factors were studied in a 3 × 4 factorial split-plot design with three replicates: *irrigation* (3) and *genotype* (4). *Irrigation* was applied to the main plot and *genotype* to sub-plot. Three irrigation treatments were applied to all genotypes of tomato by means of a drip-irrigation system: DRY (no irrigation); IRR (long-season full irrigation); REW (post-drought rewaterings). At transplant, irrigation was applied to restore field capacity (FC). To this end, the amount of water to distribute (~45 mm) was calculated, considering soil water content (measured gravimetrically on five soil samples collected randomly at 0.40 m depth within the experimental field and oven-dried at 105 °C). Thereafter, irrigation in DRY plots was stopped, following the traditional method of cultivation of these tomato landraces [9], whereas irrigation in IRR was kept until fruits ripened (mid-July), restoring 100% of water evapotranspired (ETc = ET_0_ × kc) at each watering, for a total of 9 waterings. Reference ET (ET_0_) was measured in a class-A evaporation pan; crop coefficients (kc) were those reported for tomato by Patanè et al. [27]. The amount of water to apply with irrigation (V) was calculated on the basis of maximum available soil water content (ASWC) in the first 0.4 m of soil depth [28]. Irrigation was applied when cumulative ETc matched V (~42 mm). In REW, the crop was submitted to alternating cycles of drying and rewatering. Precisely, after initial irrigation at transplant, the soil was left to dry up to wilting point (WP) (i.e., total available soil water-ASW was used by the crop); thus the crop was irrigated restoring FC at 0.4 m soil depth, and after that the soil was left to dry up again to WP until the next irrigation, for a total of two irrigations (approximately 8 weeks after transplant, the 1^st^; after further 3 weeks, the 2^nd^) of approx. 70 mm each. Total seasonal volume of water distributed by irrigation was 440, 4403, and 1646 m^3^ ha^−1^, respectively in DRY, IRR, and REW.

No chemical herbicides were used for weed control. A hand weeding was performed once only, since the crop covered the soil and weeds could no longer grow.

The crop was hand-harvested when the ripe fruits reached ~95% (late July). All fruits harvested per each plot were measured for weight altogether, and the number of plants per each plot was counted; then final fresh fruit yield per plant (g FW plant^−1^) was calculated.

### 4.2. Measurements

During the field experiment, the following meteorological variables were recorded: air temperature, rainfall, and class-A pan evaporation, using a data logger (CR10, Campbell Scientific, Logan, UT, USA) located approximately 50 m from the experimental field. From May until mid-July, soil water content was also measured for each genotype and irrigation treatment using gypsum blocks (Soilmoisture Equipment Corp., Santa Barbara, CA, USA) located at a 0.4 m soil depth in all replicates. Soil water content was measured at 2-day or 3-day intervals and always before and after each irrigation, and soil water deficit (SWD, %) was estimated according to the following formula:SWD = (1 − (WC − WP)/(FC − WP)) × 100(1)
where WC is the soil water content as a percentage of dry soil; FC is the soil water content at field capacity as a percentage of dry soil; WP is the soil water content at wilting point as a percentage of dry soil. SWD ranged between 0% (FC) and 100% (WP).

Relative water content (RWC, %) of leaves was measured weekly for total seven measurements starting from mid–late May. To this end, one leaf was removed from eight representative plants in each plot, weighed, and quickly immersed in distilled water overnight to saturation; thus, all leaves were weighed (turgor weight) after removing the excess water. After that, the leaves were oven-dried until a constant weight at 70 °C and dry weight was measured. RWC was calculated according to the following formula:RWC = (FW − DW)/(TW − DW) × 100(2)
where FW (g) is the fresh weight of leaf just removed from plant, DW (g) is the weight of leaf oven-dried at 70 °C, and TW (g) is the turgor weight of leaf after saturation.

At the same time and on the same leaves, the specific leaf area (SLA), which indicates the leaf area per unit of leaf dry weight (it may be considered as an index of leaf thickness: the lower the SLA, the greater the leaf thickness) was measured. To this end, before leaf oven drying, leaf area was measured in a leaf area meter (Delta-T, Digital Image Analysis System, Cambridge, UK). SLA (cm^2^ g^−1^), was calculated as follows:SLA = LA/DW(3)
where LA is leaf area (cm^2^) and DW is leaf dry weight at 70 °C (g).

### 4.3. Laboratory Analyses

#### 4.3.1. Leaf Sample Preparation

Together with leaf sampling for RWC measurements, further leaves (1 leaf per plant) were sampled from the same eight tomato plants. All fresh leaves were ground using a cryogenic mortar in liquid nitrogen. The powder was weighed and stored at −80 °C until analyses.

Different aliquots were used for chemical analyses. These last were carried out in triplicate.

#### 4.3.2. Proline Content Measurement

Proline content was estimated according to the method proposed by Bates et al. [29] using L-proline as standard. A one-gram sample was homogenised in 5 mL of aqueous sulfosalicylic acid (3%) and centrifuged at 14,000 *g* (Neya 10R, REMI, Mumbai, India) for 15′. Two mL of supernatant was added to 2 mL of mixed acetic acid and ninhydrin, and heated for 1 h at 100 °C. The mixture was cooled rapidly in an ice-bath, and added to 4 mL of toluene. Absorbance was read in a spectrophotometer (7315 Spectrophotometer, Jenway, Staffordshire, UK) at 525 nm. Proline content was reported as µmol g^−1^ FW.

#### 4.3.3. MDA Content Measurement

Malondialdehyde (MDA) content was measured according to Li et al. [30]. Leaf samples (0.5 g) were extracted in 1.5 mL of trichloroacetic acid (TCA) (5%) (*w/v*), then the extract was centrifuged at 5000 *g* for 10′. All surnatant was diluted to 10 mL with bi-distilled water and 2 mL of diluted extract was homogenised with 2 mL of thiobarbituric acid (TBA) (0.67%). The mixture was incubated at 95 °C for 30′, then centrifuged at 5,000 *g* for 10′. The surnatant was used for MDA estimation. Absorbance was read in a spectrophotometer (7315 Spectrophotometer, Jenway, Staffordshire, UK) at 450, 532, and 600 nm. MDA was calculated as follows:MDA (µmol L^−1^) = 6.45 × (*A*_532_ − *A*_600_) − 0.56 × *A*_450_(4)

The results were expressed as µmol g^−1^ FW.

#### 4.3.4. Extraction and Assay of Antioxidant Enzymes

To study the enzymatic response of the tomato plants against soil water deficit stress, antioxidant enzymes GPX, CAT, and SOD were analyzed in leaves. To this end, leaf samples (0.5 g) were extracted in 4 mL buffer (50 mM potassium phosphate, 1 mM ethylenediaminetetraacetic acid-EDTA, 1% polyvinylpyrrolidone-PVP, 1 mM dithiothreitol-DTT, and 1 mM phenylmethylsulfonyl-PMSF) (pH 7.8). Samples were centrifuged at 4 °C (15,000 *g* for 30′) and supernatant was collected and stored at −80 °C for the enzyme assay.

The glutathione peroxidase activity (GPX; EC 1.11.1.9) was measured according to Ruley et al. [31]. Twenty µL of extract and 17 mM H_2_O_2_ were homogenized in a 2% guaiacol solution to obtain a final volume of 1 mL. The increase in absorbance was monitored at 510 nm every 1′, for a total of 3′. The GPX activity was expressed as units µmol min^−1^ g^−1^ FW.

The catalase activity (CAT; EC 1.11.1.6) was measured according to Aebi et al. [32]. Twenty µL of extract were homogenized in 830 µL potassium phosphate buffer solution (50 mM, pH 7). To start the reaction, 150 µL of H_2_O_2_ was added, whose decrease was monitored at 240 nm, for 2′. CAT was expressed as units nmol min^−1^ g^−1^ FW.

The superoxide dismutase activity (SOD; EC 1.15.1.1) was measured according to Giannopolitis and Ries [33]. The reaction mixture containing riboflavin 0.1 mL, L-methionine 1.5 mL, NBT 1.0 mL, Triton-X 0.75 mL, EDTA 2 mM, and NaKPi 27.0 mL at pH 7.8 was added to 0.5 mL of enzyme extract, to obtain a final volume of 3.0 mL. The tubes were placed at 25 °C for 10′ in the absence of direct light. The reaction began with the exposure of the samples to a white light fluorescent lamp (15 WTS pre-heated, 6500 K). SOD activity was read at 560 nm. A SOD unit was defined as the amount of enzyme required for 50% inhibition of the reduction of NBT. The SOD activity was expressed as units-U mg^−1^ FW.

All readings were made by a spectrophotometer (7315 Spectrophotometer, Jenway, Staffordshire, UK).

### 4.4. Statistical Analyses

Data of RWC, SLA, proline, MDA, GPX, CAT, and SOD were subjected to a post-hoc three-way analysis of variance (ANOVA) using CoStat version 6.003 software (CoHort Software, Monterey, CA, USA). *Irrigation, genotype, time of measurement*, and their interaction were considered as sources of variation. *Irrigation* and *genotype* were considered as fixed factors, whereas *time of measurement* as a random factor. Data of fruit yield per plant were subjected to a two-way analysis of variance (ANOVA) using CoStat version 6.003 software (CoHort Software, Monterey, CA, USA), considering *irrigation* and *genotype* and their interaction as sources of variation. Means were separated by the Tukey’s test at 95% confidence level.

A Pearson’s correlation test was conducted among data of SWD, RWC, SLA, proline, MDA, and antioxidant enzymes, separately per tomato genotype (SigmaPlot11, Systat Software Inc., San Jose, CA, USA).

## 5. Conclusions

The present study revealed that severe soil water deficit conditions may determine minor oxidative damage in long shelf-life tomatoes, as revealed by lower levels of MDA measured with respect to the commercial tomato. The significant correlation of CAT vs. soil water deficit described in all tomatoes indicates that this antioxidant enzyme, among those analyzed, may be considered as a biomarker for a (potential) water stress condition more than for oxidative stress due to water deficit. The application of water-saving irrigation strategies through the adoption of appropriate cycles of drought and rewaterings may contribute to promote plant productivity of the low-yielding local landraces of tomato under restricted water availability in a semi-arid climate.

## Figures and Tables

**Figure 1 plants-11-03045-f001:**
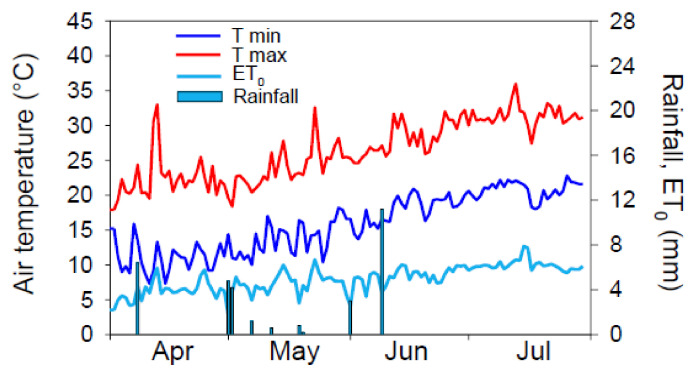
Meteorological course (minimum-Tmin and maximum-Tmax air temperature, rainfall, reference evapotranspiration-ET_0_) recorded in the experimental site from April to July 2016.

**Figure 2 plants-11-03045-f002:**
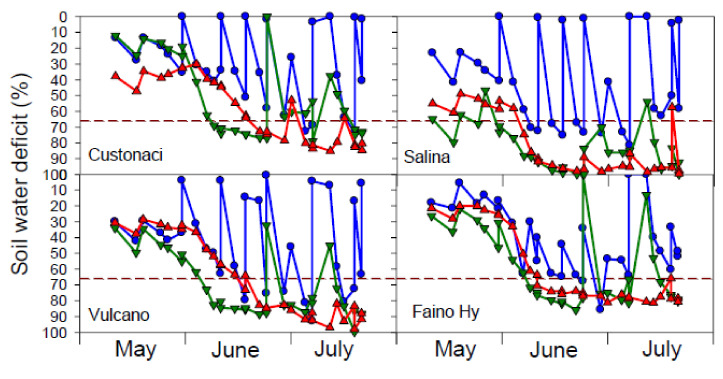
Soil water deficit measured at a depth of 0.40 m in each irrigation treatment (blue line: IRR; green line: REW; red line: DRY). Constant horizontal short dash line indicates minimum threshold of available soil water content.

**Figure 3 plants-11-03045-f003:**
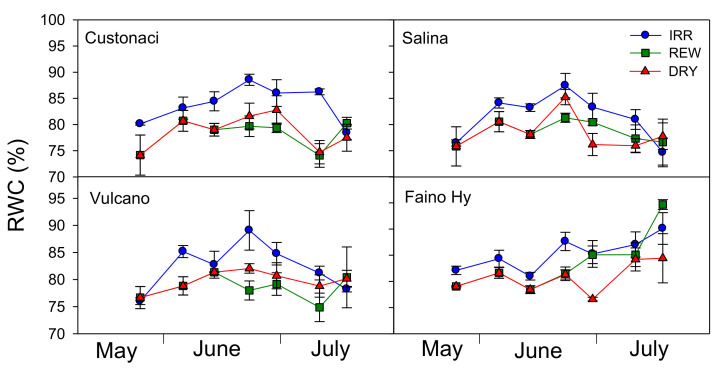
Relative water content of leaves (RWC, %) in four genotypes of tomato in relation to the irrigation treatment (IRR: full irrigation; REW: rewatering; DRY: no irrigation). Vertical bars indicate the standard error (*n* = 3).

**Figure 4 plants-11-03045-f004:**
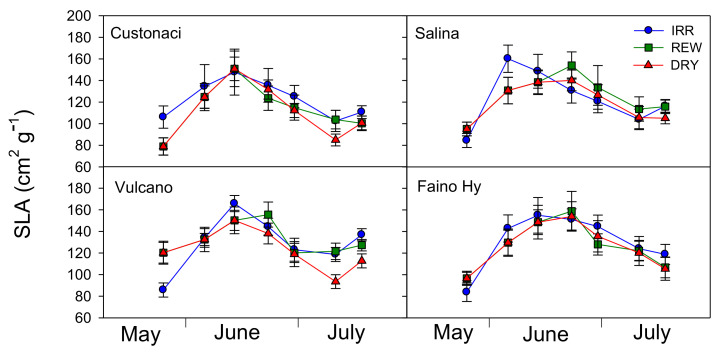
Specific leaf area (SLA, cm^2^ g^−1^) in four genotypes of tomato in relation to the irrigation treatment (IRR: full irrigation; REW: rewatering; DRY: no irrigation). Vertical bars indicate the standard error (*n* = 3).

**Figure 5 plants-11-03045-f005:**
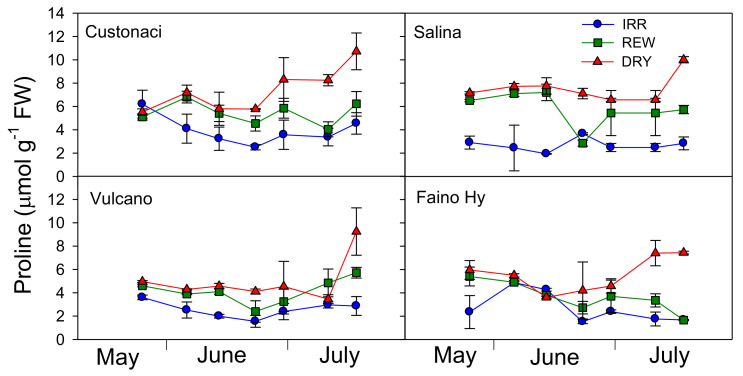
Leaf proline content (µmol g^−1^ FW) in four genotypes of tomato in relation to the irrigation treatment (IRR: full irrigation; REW: rewatering; DRY: no irrigation). Vertical bars indicate the standard error (*n* = 3).

**Figure 6 plants-11-03045-f006:**
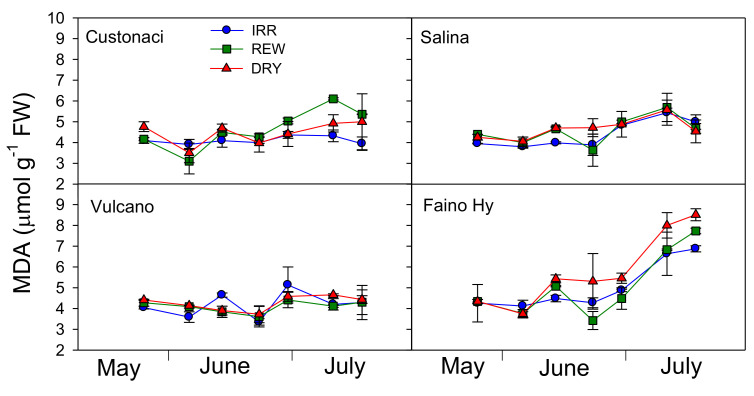
Malondialdehyde leaf content (MDA, µmol g^−1^ FW) in four genotypes of tomato in relation to the irrigation treatment (IRR: full irrigation; REW: rewatering; DRY: no irrigation). Vertical bars indicate the standard error (*n* = 3).

**Figure 7 plants-11-03045-f007:**
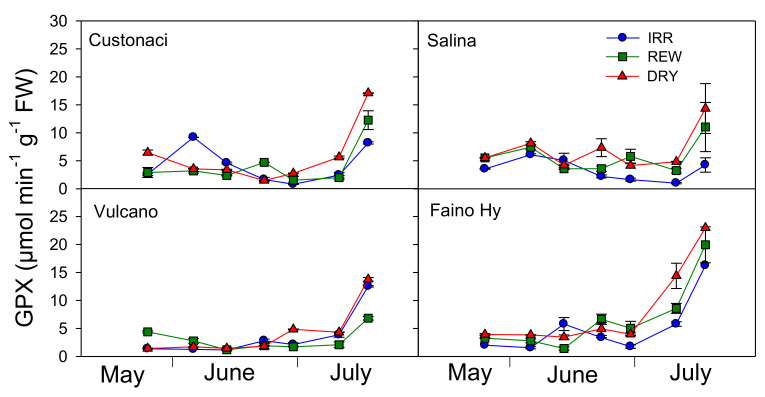
Glutathione peroxidase activity (GPX, µmol min^−1^ g^−1^ FW) in four genotypes of tomato in relation to the irrigation treatment (IRR: full irrigation; REW: rewatering; DRY: no irrigation). Vertical bars indicate the standard error (*n* = 3).

**Figure 8 plants-11-03045-f008:**
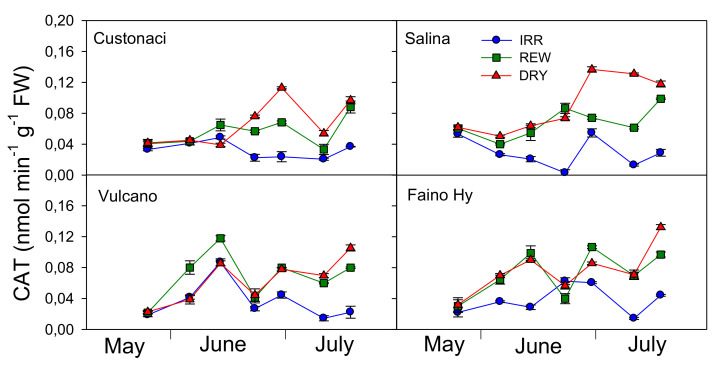
Catalase activity (CAT, nmol min^−1^ g^−1^ FW) in four genotypes of tomato in relation to the irrigation treatment (IRR: full irrigation; REW: rewatering; DRY: no irrigation). Vertical bars indicate the standard error (*n* = 3).

**Figure 9 plants-11-03045-f009:**
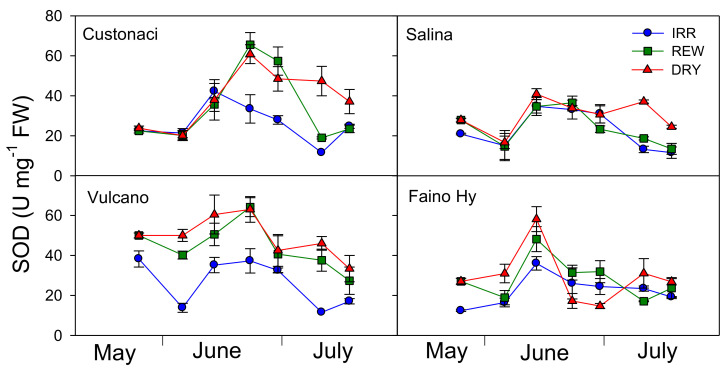
Superoxide dismutase activity (SOD, Units-U mg^−1^ FW) in four genotypes of tomato in relation to the irrigation treatment (IRR: full irrigation; REW: rewatering; DRY: no irrigation). Vertical bars indicate the standard error (*n* = 3).

**Table 1 plants-11-03045-t001:** Mean effects of *irrigation* (*I*) (IRR: full irrigation; REW: rewatering; DRY: no irrigation), *genotype* (*G*), and *time of measurement* (*T*) on RWC, SLA, and proline content in tomato. Values within each column, followed by the same letter, do not significantly differ at *p* ≤ 0.05 (Tukey’s test). ANOVA significance results are also reported: significant at *p* ≤ 0.05 (*), 0.01 (**), and 0.001 (***).

		RWC (%)	SLA (cm^2^ g^−1^)	Proline (µmol g^−1^)
Irrigation (*I*)	IRR	83.3 a	127.0 a	3.26 c
	REW	79.2 b	124.4 b	4.62 b
	DRY	79.4 b	120.8 c	6.23 a

Genotype (*G*)	Custonaci	80.2 b	116.3 c	5.28 b
	Salina	79.5 b	123.2 b	5.70 a
	Vulcano	80.3 b	128.7 a	3.91 c
	Faino Hy	82.6 a	128.0 a	3.94 c

Time of	May 26	77.2 c	95.2 e	5.26 a
Measurement (*T*)	June 6	81.8 ab	133.9 b	5.58 a
	June 15	80.4 b	149.4 a	4.59 b
	June 22	83.6 a	143.1 a	3.31 c
	June 28	81.6 ab	124.3 c	4.35 b
	July 11	80.0 b	109.5 d	4.16 b
	July 18	79.9 b	113.1 d	5.72 a

Significance	Irrigation (*I*)	***	***	***
	Genotype (*G*)	***	***	***
	Time (*T*)	***	***	***
	*I* × *G*	*	*	**
	*I* × *T*	**	***	***
	*G* × *T*	***	***	***
	*I × G × T*	*	***	***

**Table 2 plants-11-03045-t002:** Mean effects of *irrigation* (*I*) (IRR: full irrigation; REW: rewatering; DRY: no irrigation), *genotype* (*G*), and *time of measurement* (*T*) on MDA, GPX, CAT, and SOD, in tomato (on a fresh weight basis). Values within each column, followed by the same letter, do not significantly differ at *p* ≤ 0.05 (Tukey’s test). ANOVA significance results are also reported: significant at *p* ≤ 0.001 (***).

		MDA (µmol g^−1^ FW)	GPX (µmol min^−1^ g^−1^)	CAT (nmol min^−1^ g^−1^)	SOD (U mg^−1^)
Irrigation (*I*)	IRR	4.44 c	4.10 c	0.034 c	25.25 c
	REW	4.60 b	4.92 b	0.066 b	32.91 b
	DRY	4.80 a	6.27 a	0.099 a	37.10 a

Genotype (*G*)	Custonaci	4.41 b	4.70 b	0.084 a	33.48 b
	Salina	4.55 b	5.35 b	0.062 b	25.72 d
	Vulcano	4.18 c	3.59 d	0.062 b	33.48 b
	Faino Hy	5.33 a	6.75 a	0.056 b	27.73 c

Time of	26 May	4.27 cd	3.58 c	0.036 d	29.17 c
Measurement (*T*)	6 June	3.82 e	4.29 b	0.048 cd	23.20 e
	15 June	4.50 c	3.13 c	0.067 b	42.90 a
	22 June	4.02 de	3.54 c	0.106 a	41.86 a
	28 June	4.79 b	3.01 c	0.077 b	35.46 b
	11 July	5.54 a	4.85 b	0.051 c	26.16 d
	18 July	5.39 a	13.29 a	0.079 b	23.53 de

Significance	Irrigation (I)	***	***	***	***
	Genotype (G)	***	***	***	***
	Time (T)	***	***	***	***
	*I* × *G*	***	***	***	***
	*I* × *T*	***	***	***	***
	*G* × *T*	***	***	***	***
	*I* × *G* × *T*	***	***	***	***

**Table 3 plants-11-03045-t003:** Fruit yield per plant in four genotypes of tomato in relation to the irrigation treatment (IRR: full irrigation; REW: rewatering; DRY: no irrigation). For the main effects (*I* = *irrigation,* G *= genotype*), means followed by the same letter are not significantly different at *p* ≤ 0.05 (Tukey’s test). LSD interaction value is also reported.

**Irrigation** **Treatment (*I*)**		**Fruit Yield (g FW Plant^−1^)**			
	**Genotype (*G*)**				
	Custonaci	Salina	Vulcano	Faino Hy	Average (*G*)
IRR	452.5	607.4	401.5	1751.4	802.9 a
REW	383.1	465.8	315.5	1386.5	635.7 b
DRY	298.0	336.3	248.2	1103.7	496.5 c
Average (*I*)	377.9 bc	469.8 b	321.7 c	1413.6 a	
LSD*_I_*_×_*_G_*_(0.05)_	172.7				

**Table 4 plants-11-03045-t004:** Pearson’s correlation test among traits (genotype: ‘Custonaci’). Significant at *p* ≤ 0.05 (*), 0.01 (**), and 0.001 (***); not significant (ns).

	RWC	SLA	Proline	MDA	GPX	CAT	SOD
**SWD**	0.343 ^ns^	0.001 ^ns^	0.711 ***	0.303 ^ns^	0.309 ^ns^	0.663 **	0.284 ^ns^
**RWC**	-	0.570 **	−0.440 *	0.423 ^ns^	−0.246 ^ns^	−0.167 ^ns^	−0.001 ^ns^
**SLA**	-	-	−0.323 ^ns^	−0.354 ^ns^	−0.272 ^ns^	−0.065 ^ns^	0.253 ^ns^
**Proline**	-	-	-	0.119 ^ns^	0.484 *	0.703 ***	0.199 ^ns^
**MDA**	-	-	-	-	0.222 ^ns^	0.240 ^ns^	0.086 ^ns^
**GPX**	-	-	-	-	-	0.426 ^ns^	−0.110 ^ns^
**CAT**	-	-	-	-	-	-	0.517 *

**Table 5 plants-11-03045-t005:** Pearson’s correlation test among the studied traits (genotype: ‘Salina’). Significant at *p* ≤ 0.05 (*), 0.01 (**), and 0.001 (***); not significant (ns).

	RWC	SLA	Proline	MDA	GPX	CAT	SOD
**SWD**	−0.124 ^ns^	0.359 ^ns^	0.612 **	−0.047 ^ns^	0.530 *	0.671 ***	0.237 ^ns^
**RWC**	-	0.620 **	−0.357 ^ns^	−0.387 ^ns^	−0.168 ^ns^	−0.496 *	0.140 ^ns^
**SLA**	-	-	−0.181 ^ns^	−0.354 ^ns^	−0.037 ^ns^	−0.184 ^ns^	0.158 ^ns^
**Proline**	-	-	-	−0.157 ^ns^	0.617 **	0.556 **	0.123 ^ns^
**MDA**	-	-	-	-	−0.168 ^ns^	0.316 ^ns^	−0.099 ^ns^
**GPX**	-	-	-	-	-	0.431 ^ns^	−0.287 ^ns^
**CAT**	-	-	-	-	-	-	0.217 ^ns^

**Table 6 plants-11-03045-t006:** Pearson’s correlation test among the studied traits (genotype: ‘Vulcano’). Significant at *p* ≤ 0.05 (*), and 0.001 (***); not significant (ns).

	RWC	SLA	Proline	MDA	GPX	CAT	SOD
**SWD**	0.116 ^ns^	0.124 ^ns^	0.507 *	0.279 ^ns^	0.064 ^ns^	0.794 ***	0.327 ^ns^
**RWC**	-	0.498 *	−0.386 ^ns^	−0.212 ^ns^	−0.109 ^ns^	0.084 ^ns^	−0.237 ^ns^
**SLA**	-	-	0.327 ^ns^	−0.376 ^ns^	−0.206 ^ns^	0.233 ^ns^	0.207 ^ns^
**Proline**	-	-	-	0.185 ^ns^	0.533 *	0.418 ^ns^	0.120 ^ns^
**MDA**	-	-	-	-	0.242 ^ns^	0.174 ^ns^	−0.197 ^ns^
**GPX**	-	-	-	-	-	0.105 ^ns^	−0.392 ^ns^
**CAT**	-	-	-	-	-	-	0.191 ^ns^

**Table 7 plants-11-03045-t007:** Pearson’s correlation test among the studied traits (genotype: ‘Faino Hy’). Significant at *p* ≤ 0.05 (*), 0.01 (**), and 0.001 (***); not significant (ns).

	RWC	SLA	Proline	MDA	GPX	CAT	SOD
**SWD**	0.102 ^ns^	0.239 ^ns^	0.418 ^ns^	0.257 ^ns^	0.199 ^ns^	0.697 ***	0.290 ^ns^
**RWC**	-	0.172 ^ns^	−0.489 *	0.490 *	0.593 **	0.074 ^ns^	−0.306 ^ns^
**SLA**	-	-	−0.201 ^ns^	−0.311 ^ns^	−0.341 ^ns^	0.075 ^ns^	0.362 ^ns^
**Proline**	-	-	-	0.120 ^ns^	0.145 ^ns^	0.262 ^ns^	0.111 ^ns^
**MDA**	-	-	-	-	0.839 ***	0.497 *	−0.049 ^ns^
**GPX**	-	-	-	-	-	0.429 ^ns^	−0.092 ^ns^
**CAT**	-	-	-	-	-	-	0.328 ^ns^

## Data Availability

Not applicable.

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
