# Peer review of "Relative Water Content, Proline, and Antioxidant Enzymes in Leaves of Long Shelf-Life Tomatoes under Drought Stress and Rewatering"

_plants, 2022, doi:10.3390/plants11223045_

Round 1

Reviewer 1 Report

Dear autors,

I have been finished to review the manuscript entitled:

„Relative water content, proline, and antioxidant enzymes in leaves of long shelf-life tomatoes under drought stress and re-watering“

The information content of the manuscript is very interesting and popular, particularly due to  global climate change and local varieties that may carry useful traits for enhancing the tolerance to abiotic stresses through breeding.

Though MS is well written, it needs major revisions to be more focused on objectives of the MS.

First of all a field experiment with treatments should be tested at least for 2 years, because this is an uncontrolled condition. It is not common for field trials to be conducted for only one year, especially since the fruit yield was also monitored during the research. Therefore if authors have data from second year they should be included in the MS.

Besides a small number of exceptions (which I have commented in the annotated MS), I found the introduction, material and method generally appropriate, unlike the statistical treatment of data, interpretation of the results and presentation of obtained results.

The goal of this study was the identification of tolerant tomato genotypes to drought.

Authors wrote in summary: „Some physiological, oxidative, and antioxidant enzymatic patterns, were assessed in plants of three local Sicilian landraces of long shelf-life tomatoes (‘Custonaci’, ‘Salina’ and ‘Vul-13 cano’), as compared to the commercial tomato hybrid ‘Faino’ (control)“. However, in all Figures they showed differences between different treatments per cultivar. Figures should be presented the difference between the investigated autochthonous cultivars and compared them to commercial. It is not clear from the figures if there is a difference between the cultivars or not? Is it significantly different or not?

Regarding statistical analyses they wrote that data of fruit production per plant were subjected to a split-plot analysis of variance (ANOVA) considering irrigation and genotype and their interaction as sources of variation. Missing statistical analyses for RWC, proline, SLA, MDA, GPX, CAT, SOD. It is difficult to follow the results in Figures 3-9, since they only have standard errors? The results should be presented with a post hoc test to see if there is a statistically significant difference between the cultivars in different treatments.

Discussion section needs to be more comprehensive by adhering to objectives and avoiding repeating of results (particularly antioxidant enzymes). In the discussion, all results related to this study should be discussed and compared with more other similar researches. (not enough references).

Author Response

October 26, 2022

We deeply thank the reviewer for his careful reading of the manuscript and his constructive remarks. We have taken all referee’s comments into consideration to improve the manuscript. 

Please find below a detailed point-by-point response to all comments (reviewer’s comments in black, our replies in red). All corrections are reported in red throughout the manuscript.

Response to referee #1

The information content of the manuscript is very interesting and popular, particularly due to global climate change and local varieties that may carry useful traits for enhancing the tolerance to abiotic stresses through breeding.

Though MS is well written, it needs major revisions to be more focused on objectives of the MS.

First of all a field experiment with treatments should be tested at least for 2 years, because this is an uncontrolled condition. It is not common for field trials to be conducted for only one year, especially since the fruit yield was also monitored during the research. Therefore, if authors have data from second year they should be included in the MS.

The sentence of the referee could generally be considered correct. However, this experiment somehow follows the study started few years ago on these tomato landraces, where physiological traits like proline content, RWC, together with instantaneous water use efficiency, leaf transpiration, and their relationships with fruit yield, have been assessed, under the same climatic conditions. We may say that final fruit yield obtained in this study corroborates that obtained in our previous studies. Moreover, the experiment was conducted in summertime under irrigation regime, when rain is negligible or totally absent, at least in the experimental site of the present research, therefore the climatic conditions occurring during this period in this environment do not much vary along the years.

Besides a small number of exceptions (which I have commented in the annotated MS), I found the introduction, material and method generally appropriate, unlike the statistical treatment of data, interpretation of the results and presentation of obtained results.

The goal of this study was the identification of tolerant tomato genotypes to drought.

Authors wrote in summary: „Some physiological, oxidative, and antioxidant enzymatic patterns, were assessed in plants of three local Sicilian landraces of long shelf-life tomatoes (‘Custonaci’, ‘Salina’ and ‘Vul-13 cano’), as compared to the commercial tomato hybrid ‘Faino’ (control)“. However, in all Figures they showed differences between different treatments per cultivar. Figures should be presented the difference between the investigated autochthonous cultivars and compared them to commercial. It is not clear from the figures if there is a difference between the cultivars or not? Is it significantly different or not?

We have included data of each trait into a single figure, one for genotype instead of one for irrigation treatment, because we wanted to highlighted the response of each genotype to REWATERING as compared to DRY (as also highlighted in the manuscript title), to figure out if few rewaterings may affect the physiological response of these tomatoes, as compared to a no-irrigation condition, which is the traditional method of cultivation of these landraces, as compared to a commercial hybrid. To emphasize the differences among irrigation treatments, among genotypes, and among genotypes within the irrigation treatments, according to referee’s suggestion we made a post-hoc statistical analysis for RWC, proline, SLA, MDA, GPX, CAT, SOD, considering three experimental factors: irrigation, genotype, time of measurement. For readers’ convenience, we included two new tables (one for RWC, SLA, and proline, one for MDA, GPX, CAT, and SOD) into the manuscript, with the results of the 3-way statistical analysis and the mean effects ‘irrigation’, ‘genotype’, ‘time of measurements’, with relative mean separation according to the Tukey’s test.

Regarding statistical analyses they wrote that data of fruit production per plant were subjected to a split-plot analysis of variance (ANOVA) considering irrigation and genotype and their interaction as sources of variation. Missing statistical analyses for RWC, proline, SLA, MDA, GPX, CAT, SOD. It is difficult to follow the results in Figures 3-9, since they only have standard errors? The results should be presented with a post hoc test to see if there is a statistically significant difference between the cultivars in different treatments.

As above mentioned, we made a post-hoc statistical analysis for RWC, proline, SLA, MDA, GPX, CAT, SOD, considering three experimental factors: irrigation, genotype, time of measurement, and included two new tables reporting the results of the 3-way statistical analysis and the mean effects ‘irrigation’, ‘genotype’, ‘time of measurements’, with relative mean separation according to the Tukey’s test.

Discussion section needs to be more comprehensive by adhering to objectives and avoiding repeating of results (particularly antioxidant enzymes). In the discussion, all results related to this study should be discussed and compared with more other similar researches (not enough references).

The Discussion section has been re-modulated, eliminating some repetitions and including new references.

I suggest to present fruit yield in Table 1 separately by every irrigation treatment, for all cultivars together. 

Table 1 (now Table 3) is comprehensive of all statistical information. Indeed, it reports the effects of Irrigation, Genotype, and their interaction, on final fruit yield per plant. Therefore, data of fruit yield separately per each irrigation treatment, on average of cultivars, is already reported.

Reviewer 2 Report

Title: Relative water content, proline, and antioxidant enzymes in leaves of long shelf-life tomatoes under drought stress and re-watering

 The experiment is done well and the theme is novel and very interesting, but some improvements are needed. In general, the manuscript may meet the publication standard of the journal after revisions.

 Introduction:

- Line 34: “O2 is oxygen molecule not superoxide anion, please change it to “O2‒”.

- Line 67: please correct the [9,11÷14].

- Please support the introduction and discussion with the following articles by cite them in appropriate places:

(1) Alharby, H.F., Alzahrani, H.S., Alzahrani, Y., Alsamadany, H., Hakeem, K.R., Rady, M.M. (2021). Maize grain extract enriched with polyamines alleviates drought stress in Triticum aestivum through up-regulation of the ascorbate-glutathione cycle, glyoxalase system, and polyamine gene expression. Agronomy, 11(5): 949.

(2) Rady, M.M., Boriek, S.H.K., Abd El-Mageed, T.A., Seif El-Yazal, M.A., Ali, E.F., Hassan, F.A.S., Abdelkhalik, A. (2021). Exogenous Gibberellic Acid or Dilute Bee Honey Boosts Drought Stress Tolerance in Vicia faba by Rebalancing Osmoprotectants, Antioxidants, Nutrients, and Phytohormones. Plants - MDPI, 10(4): 748.

(3) Desoky, E.S., Mansour, E., Ali, M.M.A., Yasin, M.A.T., Abdul-Hamid, M.I.E., Rady, M.M., Ali E.F. (2021). Exogenously used 24-epibrassinolide promotes drought tolerance in maize hybrids by improving plant and water productivity in an arid environment. Plants - MDPI, 10(2): 354.

(4) Rady, M.M., Belal, H.E.E., Gadallah, F.M., Semida, W.M. (2020). Selenium application in two methods elevates drought tolerance in Solanum lycopersicum by increasing yield, quality, and antioxidant defense system and suppressing oxidative stress biomarkers. Scientia Horticulturae, 266: 109290.

 Conclusions:

- Please summarize your conclusions to 3-4 sentences in just one paragraph.

Author Response

We deeply thank the reviewer for his careful reading of the manuscript and his constructive remarks. We have taken all referee’s comments into consideration to improve the manuscript. 

Please find below a detailed point-by-point response to all comments (reviewer’s comments in black, our replies in red). All corrections are reported in red throughout the manuscript.

 Response to referee #2

The experiment is done well and the theme is novel and very interesting, but some improvements are needed. In general, the manuscript may meet the publication standard of the journal after revisions.

Introduction:

Line 34: “O2” is oxygen molecule not superoxide anion, please change it to “O2•â€’”.

The correction has been made according to referee’s suggestion.

Line 67: please correct the [9,11÷14].

The correction has been made in the text.

Please support the introduction and discussion with the following articles by cite them in appropriate places:

(1) Alharby, H.F., Alzahrani, H.S., Alzahrani, Y., Alsamadany, H., Hakeem, K.R., Rady, M.M. (2021). Maize grain extract enriched with polyamines alleviates drought stress in Triticum aestivum through up-regulation of the ascorbate-glutathione cycle, glyoxalase system, and polyamine gene expression. Agronomy, 11(5): 949.

(2) Rady, M.M., Boriek, S.H.K., Abd El-Mageed, T.A., Seif El-Yazal, M.A., Ali, E.F., Hassan, F.A.S., Abdelkhalik, A. (2021). Exogenous Gibberellic Acid or Dilute Bee Honey Boosts Drought Stress Tolerance in Vicia faba by Rebalancing Osmoprotectants, Antioxidants, Nutrients, and Phytohormones. Plants - MDPI, 10(4): 748.

(3) Desoky, E.S., Mansour, E., Ali, M.M.A., Yasin, M.A.T., Abdul-Hamid, M.I.E., Rady, M.M., Ali E.F. (2021). Exogenously used 24-epibrassinolide promotes drought tolerance in maize hybrids by improving plant and water productivity in an arid environment. Plants - MDPI, 10(2): 354.

(4) Rady, M.M., Belal, H.E.E., Gadallah, F.M., Semida, W.M. (2020). Selenium application in two methods elevates drought tolerance in Solanum lycopersicum by increasing yield, quality, and antioxidant defense system and suppressing oxidative stress biomarkers. Scientia Horticulturae, 266: 109290.

Three of the above suggested articles have been included in the text and in list of references.

Conclusions:

Please summarize your conclusions to 3-4 sentences in just one paragraph.

We have summarized the Conclusion section to 3 sentences in one paragraph, as follows:

‘The present study revealed that severe soil water deficit conditions may determine a minor oxidative damage in long shelf-life tomatoes, as revealed by lower levels of MDA measured respect to the commercial tomato. Significant correlation of CAT vs. soil water deficit described in all tomatoes indicates that this antioxidant enzyme, among those analyzed, may be considered as a biomarker for a (potential) water stressing condition more than for an oxidative stress due to water deficit. The application of water saving irrigation strategies through the adoption of appropriate cycles of drought and rewaterings may contribute to promote plant productivity of the low-yielding local landraces of tomato under restricted water availability in semi-arid climate.’

Round 2

Reviewer 1 Report

In materials and methods sections, the fruit yield part is missing. It is unclear what was measured? Please explain whether this is the total number of fruits per plant, fruit weight per plant (kg or g/plant)?

Author Response

November 7, 2022

To:

Ms. Jaine Wang

Assistant Editor

Plants MDPI

Object:

Revision of Ms. plants- 1993593-R1  ‘Relative water content, proline, and antioxidant enzymes in leaves of long shelf-life tomatoes under drought stress and rewatering

Authors:

Cristina Patanè, Salvatore Luciano Cosentino, Daniela Romano, Stefania Toscano

Dear Assistant Editor,

We deeply thank again the reviewer for his careful reading of the manuscript and his constructive remarks. We have taken all referee’s comments into consideration to improve the manuscript. 

Please find below a detailed point-by-point response to all comments (reviewer’s comments in black, our replies in red). All corrections are reported in red throughout the manuscript.

Response to referee #1

The authors have made all the corrections suggested by both reviewers. Only a small correction is missing (came from the last round of revision of referee 1, see below):

In materials and methods sections, the fruit yield part is missing. It is unclear what was measured? Please explain whether this is the total number of fruits per plant, fruit weight per plant (kg or g/plant)?

The reviewer is referring to the results presented in section "Fruit production".

In addition, please just check the numbering of the subsection 2 (Results). The authors begin with 2.1 and then put 3.2, 3.3...

The authors should just clarify the fruit yield calculation and correct the subsection numbering before the acceptance

The explanation for fruit yield calculation was preliminary included at the end of paragraph 4.2 Measurements. Now we have moved and elucidated this explanation at the end of paragraph 4.1 Open-field experiment.

As discussed in the ‘Results’ section (see paragraph 3.8), what is reported is total fruit yield per plant as g plant-1 (as also reported in table 3). According to referee suggestion in the 1st round of revision, ‘Fruit production’ was corrected in ‘Fruit’ yield’, keeping the same unit (g plant-1). We have now corrected the heading for 3.8 Fruit production in 3.8 Fruit yield per plant.  

WE have clarified fruit harvest in Mat and Met section, as follows:

The crop was hand harvested when the ripe fruits reached ~95% (late July). All fruits harvested per each plot were measured for weight altogether, and the number of plants per each plot was counted, then final fresh fruit yield per plant (g FW plant-1) was calculated’.

We have checked and corrected all subsections numbering.
